# Dynamic Strength Characteristics of Cement-Improved Silty Clay under the Effect of Freeze-Thaw Cycles

**Zheng Ma [1], Zhen Xing [1], Yingying Zhao [2,\*] and Yiru Hu [3]**

1   Xingtai Construction Group Co., Ltd., Hohhot 010000, China
2   School of Civil Engineering, Qingdao University of Technology, Qingdao 266033, China
3   School of Civil Engineering, Harbin Institute of Technology, Harbin 150090, China
\*   Correspondence: zhaoyingying@qut.edu.cn

**Abstract:** In the seasonally frozen soil regions of northern China, silty clay is widely used as a subgrade bed filler in heavy-haul railway construction. In this paper, the influence of freeze-thaw cycles on the dynamic strength properties (strength parameters and dynamic critical stress) of silty clay fillers before and after cement improvement was investigated by a series of dynamic triaxial tests under different confining pressure conditions, and the test results were quantified to analyze the improvement effects of cement improvement. The results show that cement modification can significantly improve the dynamic strength parameters (dynamic strength, dynamic strength index, and critical dynamic stress) of silty clay before and after freezing and thawing. The dynamic strength of cement-improved silty clay (CSC) was improved by 2.8 to 5.2 times compared to silty clay, and a high level of dynamic strength can be maintained after multiple freeze-thaw cycles. The dynamic cohesion was increased by 1.5 to 3 times and the dynamic internal friction angle was increased by 1.5 to 4 times. The attenuation rate of the critical dynamic stress of CSC with the number of freeze-thaw cycles was greater than that of the plain filler, while the relative lifting effect of the critical dynamic stress of the cement improvement was significant after three freeze-thaw cycles, and the maximum value was reached at a cycle number of three, with a relative increase of 2.5 times. A new index of critical dynamic stress attenuation of CSC for freeze-thaw cycles was introduced, which provides a useful reference for subgrade improvement and reinforcement along the silty clay railway in northern China.

**Keywords:** silty clay; cement improvement; freeze-thaw cycle; dynamic strength characteristics

## 1. Introduction

In northern China, seasonally frozen soil is widely distributed on major railway lines and branch lines. Under the coupling effect of the freeze-thaw cycle and train dynamic load, the vibration subsidence disease occurs year after year. Although the railway department has invested a lot of funds to control it every year, it have had little success. Especially for the Beijing–Harbin Railway, the roadbed filler is mostly silty clay, which belongs to C-group soil. If reinforcement and improvement are not carried out reasonably, such vibration and subsidence disease will be increasingly aggravated with the heavy-haul train formation and the continuous increase in axle load of vehicles with traction weight.

As a kind of soil improvement material with good performance and relatively low cost, cement has been widely used as the underlying filler of subgrade foundations in China, such as Datong-Xi'an high-speed railway [1], Beijing–Harbin high-speed railway [2], Qinhuangdao-Shenyang high-speed railway [3], XilinHot-WulanHot railway [4], etc. Domestic and foreign scholars have conducted a great deal of research on the mechanical properties of cement-improved filler through indoor and outdoor tests and verified its applicability and effectiveness as a subgrade filler [5–8]. In view of the mechanical properties of cement-improved soils after freeze-thaw cycles, the research mainly focused on static

properties. Most of the studies showed that cement amendment could inhibit the volume change of clays under freeze-thaw cycles [9], and the parameters, such as compressive strength and elastic modulus of cement-improved soil were maintained at a high level after several freeze-thaw cycles [10] and were significantly better than lime-improved soil [11]. The static triaxial test conducted by Wang Tianliang et al. on cement clay showed that the shear strength of cement soils increased significantly, the stress-strain relationship was the strain-softening type, and the brittleness failure was the main failure [12]. The strength decline rate after the freeze-thaw cycles was lower than that of plain filler, and the cohesion decreased with the increase in the number of freeze-thaw cycles, and there exists an optimum dosage of cement-improved loss under freeze-thaw cycles [13]. With regard to the dynamic stability of cement-improved soils under freeze-thaw cycles, Liu Jiankun and Wang Tianliang showed that cement can effectively increase the critical dynamic stress of clay soils through indoor freeze-thaw cycles and dynamic triaxial tests, with the increase in freeze-thaw times, the attenuation rate of the critical dynamic stress of cement-improved soil is lower than that of plain filler [14–16]. There is a critical value of resilient modulus of cement-improved soil, which is proportional to freezing temperature.

The cement-improved silty clays widely used in the subgrade of Heilongjiang railway. As a result, a great number of freeze-thaw tests and direct shear tests under various conditions have been carried out [17]. A comparison of the influences of the freeze-thaw action on lime-stabilized basalt and polypropylene fiber-reinforced clay was investigated [18]. Against this backdrop, a convenient method was proposed to evaluate the dynamic response of subgrade in a laboratory and provided a reference for the dynamic characteristics of silty clay (SC) modified using fly ash and crumb rubber(RFS) [19]. To achieve the purposes of disposing industry solid wastes and enhancing the sustainability of subgrade life-cycle service performance in seasonally frozen regions compared to previous research on modified silty clay (MSC) composed of oil shale ash (OSA), fly ash (FA), and silty clay (SC), Wei et al., 2019 identified for the first time the axial deformation characteristics of MSC with different levels of cycle load number, dynamic stress ratio, confining pressure, loading frequency, and F-T cycles; and corresponding to the above conditions, the normalized and logarithmic models on the plastic cumulative strain prediction of MSC were established [20]. Other influential work includes [21–25]. Moreover, research in this area is rare.

In this paper, the variation law of the dynamic strength of cement-modified silty clay under freeze-thaw cycles was studied using the dynamic triaxial test. The enhancement effects of cement improvement on the dynamic strength parameters and critical dynamic stress of the silty clay filler under a different number of freeze-thaw cycles and confining pressure conditions were quantified and analyzed, and a new index of critical dynamic stress attenuation of cement-improved silty clay for freeze-thaw cycles was introduced, which provides a useful reference for the improvement and reinforcement of the subgrade along the silty clay railway in northern China.

## 2. Experimental Scheme

### 2.1. Experimental Material

The test soil was taken from a soil material landfill in Beijing–Harbin high-speed railway and the tested soil was air-dried and crushed using a rubber pestle. In order to avoid the influence of individual coarse grains mixed in the filler on the stability of the test results, the fillers were sieved through a 2 mm sieve. The filler was silty clay, which belongs to C-group soil, and its physical properties are shown in Table 1. The cement used for filler improvement was type A slag silicate cement (PSA) with a strength grade of 32.5 and a slag content of 33%, its slag admixture was 33%, the initial setting time was about 3.5 h, and the final setting time was about 8 h.

**Table 1.** The physical characteristics of soil sample.

| Parameter | Value | | |
|---|---|---|---|
| granulometric composition/% | 2~0.075 mm | 0.075~0.005 mm | <0.005 mm |
| | 31 | 57 | 12 |
| maximum dry density $\rho_d$/g/cm$^3$ | 1.65 | | |
| Liquid limit $\omega_L$/% | 34.1 | | |
| plastic limit $\omega_P$/% | 18.2 | | |
| Plasticity Index $I_P$ | 15.9 | | |
| Optimum moisture content/% | 16.7 | | |

### 2.2. Preparation of Specimens

The optimum moisture content of the modified cement clay is similar to that of plain filler; therefore, 16.7 + 3% distilled water was added to the soil using the spray method after drying, and 3% of water was used for the chemical reaction between the cement and the minerals in the soil particles. The soil was mixed well, sealed and left for 24 h. Cement was added to the soil at a blending rate of 6%, stirred evenly, and samples were prepared within the initial setting time of cement, wherein the blending rate was the percentage of the ratio of cement mass to soil mass. The maximum dry density of cement-improved soil was 1.71 g/cm$^3$ using the compaction test, and the compaction coefficient of 0.95 was taken to control the dry density of plain filler and cement-improved soil during sample preparation. The sample was prepared using the batch pressing method. In order to reduce the error between samples, under the same working condition, the samples were as identical as possible to avoid the influence of external factors. The sample was compacted into three layers using a prototype and a saturator. The height of the sample was 125 mm and the diameter was 61.8 mm. The cement-improved soil sample was put into the humidor and cured for 28 days before the test.

### 2.3. Test Method

The prepared sample was sealed with a plastic film to prevent water loss during freezing and thawing and then placed in a constant temperature test chamber for freezing and thawing tests under the condition of a closed system (i.e., no water replenishment). According to the local temperature change of the subgrade, the minimum temperature of the freezing process was set at $-15$ °C and the freezing time was 12 h, after that, the specimen was rapidly thawed at 5 °C for 12 h, which was a freezing and thawing cycle. The freeze-thaw cycle process was repeated several times, and the freeze-thaw cycle of the samples was set at 0, 1, 3, 6, and 10 cycles. The samples were taken out for dynamic triaxial tests after reaching the preset freeze-thaw cycle.

The dynamic triaxial test adopted the consolidated undrained (CU) mode, and the testing machine was MTS-810, which was made in the United States. The prepared specimens were placed in the dynamic triaxial testing machine and consolidated under equal pressure for 2 h. The consolidation envelope pressure was taken as 30, 60, and 100 kPa, which basically covered the confining level within the depth of the subgrade of the heavy-loaded road. The dynamic load was applied using the stress control method, and the loading method was a wave asymmetric sinusoidal wave.

The dynamic stress time history can be obtained from Equation (1):

$$\sigma_d(t) = \frac{\overline{\sigma}_d}{2} \sin\left(2\pi f \cdot t - \frac{\pi}{2}\right) + \frac{\overline{\sigma}_d}{2} \tag{1}$$

where $\overline{\sigma}_d$ is the dynamic stress amplitude and $f$ is the loading frequency.

In order to determine the dynamic strength and critical dynamic stress of the soil, the $\overline{\sigma}_d$ applied to multiple specimens under the same working condition should be increased step-by-step. It is assumed that the increment of each step was $\Delta\overline{\sigma}_d$, the value of $\Delta\overline{\sigma}_d$ of the plain filler was between 20 and 40 kPa, and the cement-improved soil was between 100 and

400 kPa. The loading frequency of the test was 1 Hz. For the plain filler specimens, which were dominated by plastic failure, the failure criterion was set at when the cumulative axial plastic strain of the specimen reached 5%. For the specimens of cement-improved soil, the point where the accumulated plastic strain $\varepsilon_p$-vibration number $n$ relationship curve shows a clear inflection point is used as the failure point to determine the failure vibration number. The test was terminated after 10,000 cycles of dynamic load or when the failure criteria were met.

## 3. Analysis of Experimental Results

To facilitate the discussion of the effect of cement improvement on the dynamic strength parameters and freeze-thaw resistance of plain soils, the relative improvement of cement improvement was defined as the ratio of dynamic strength parameters of cement-improved soils to plain filler under the same conditions minus 1, and the absolute improvement was defined as the difference between the dynamic strength parameters of cement-improved soils and plain filler under the same conditions. In the following, $\Delta R^i_{df}$, $\Delta R^i_C$, $\Delta R^i_\varphi$, and $\Delta R^i_{dcr}$ represent the relative increase in dynamic strength $\sigma_{df}$, dynamic cohesion $C_d$, dynamic internal friction angle $\varphi_d$, and critical dynamic stress $\sigma_{dcr}$ of cement-improved plain filler, respectively; $\Delta \sigma^i_{df}$, $\Delta C^i_d$, $\Delta \varphi^i_d$, and $\Delta \sigma^i_{dcr}$ represent the absolute improvement of the above parameters of plain filler by cement improvement, where $i$ represents the number of freeze-thaw cycles.

### 3.1. Dynamic Strength

The dynamic stress of subgrade soil under the dynamic load of train was one-way cyclic stress. In order to directly reflect the strength characteristics of soil under the dynamic load of train, this paper took the amplitude of dynamic stress $\overline{\sigma}_d$ required for the destruction of soil under repeated dynamic load as the dynamic strength of soil $\sigma_{df}$. Figure 1 shows the dynamic strength curve of plain filler and cement-improved soil when $N_{ft} = 0$, 1, and 3. It can be seen from the figure that the dynamic strength of plain filler and cement-improved soil decreases significantly after freeze-thaw cycles, and decreased continuously with the increase in dynamic loading number $n_f$. It is mainly due to the meso-fabric of geomaterials which has a significant influence on its strength, where the increase in dynamic loading number destroys the meso-fabric of the plain filler and cement-improved soil [26,27]. The dynamic strength curve of the plain filler remains basically parallel before and after the freeze-thaw cycles, which indicates that freeze-thaw cycles have little effect on the decay rate of plain filler $\sigma_{df}$ with $\log n_f$. However, the dynamic strength line of cement-improved soil shows a flattening trend with the increase in $N_{ft}$, indicating that a high dynamic strength level can still be maintained after multiple freeze-thaw cycles.

The calculation shows that the relative improvement of cement improvement for dynamic strength $\Delta R^i_{df}$ varies in the range of 3.7~6.1. Under the same freeze-thaw cycle conditions, the change of the confining pressure from 30 to 100 kPa has a very limited effect on $\Delta R^i_{df}$, indicating that the relative enhancement effect of the cement improvement on the dynamic strength of the soil tends to be stable within the influence depth of train dynamic load. For the sake of convenience, the average value of $\Delta R^i_{df}$ under each confining pressure is used to reflect the relative increase in dynamic strength under a freeze-thaw cycle, as shown in Figure 2. It can be seen that $\Delta R^i_{df}$ does not show a significant pattern with respect to $N_{ft}$. When $N_{ft} < 3$, $\Delta R^i_{df}$ decreases with the increase in $n_f$, whereas, when $N_{ft} \geq 3$, $\Delta R^i_{df}$ increases with the increase in $n_f$. This shows that when $N_{ft} < 3$, the cement improvement has a greater effect on the relative increase in dynamic strength under low cycle fatigue conditions, while for heavy haul railway subgrade filler requiring a high-cycle fatigue design ($n_f \rightarrow \infty$), the cement improvement has a more significant effect on the relative increase in dynamic strength after multiple freeze-thaw cycles ($N_{ft} \geq 3$).

Figure 3 shows the relationship between the absolute increase in dynamic strength $\Delta \sigma^i_{df}$ of cement-improved soil at $n_f = 10$, 100, and 1000 and the number of freeze-thaw cycles $N_{ft}$. The curve in the figure is the mean value of $\Delta \sigma^i_{df}$ under each confining pressure

to reflect the change rule of $\Delta\sigma_{df}^i$ with $N_{ft}$. The upper and lower bounds of the error bars are the values of $\Delta\sigma_{df}^i$ when the confining pressure was 30 kPa and 100 kPa, respectively, which are used to describe the influence of confining pressure on $\Delta\sigma_{df}^i$. As can be seen from the figure, the value of $\Delta\sigma_{df}^i$ decelerates and decreases with the increase in $n_f$, and the rate of decline decreases with the increase in the $n_f$, and finally tends to be stable after $N_{ft} = 6$. At the same time, $\Delta\sigma_{df}^i$ decreases with the increase in $n_f$. Subgrade filler for the heavy-haul railway with a high-cyclic fatigue design, the above-mentioned characteristics are conducive to maintaining the absolute improvement effect in cement improvement on dynamic strength. Compared with freeze-thaw cycle, the confining pressure within the depth of the subgrade bed has little effect on $\Delta\sigma_{df}^i$. The influence of confining pressure on $\Delta\sigma_{df}^i$ is most significant without freeze-thaw cycles, and the influence of confining pressure is weakened with the increase in $N_{ft}$. At the same time, the influence degree decreases with the increase in $n_f$, and the dynamic strength decreases with the increase in the number of freeze-thaw cycles.

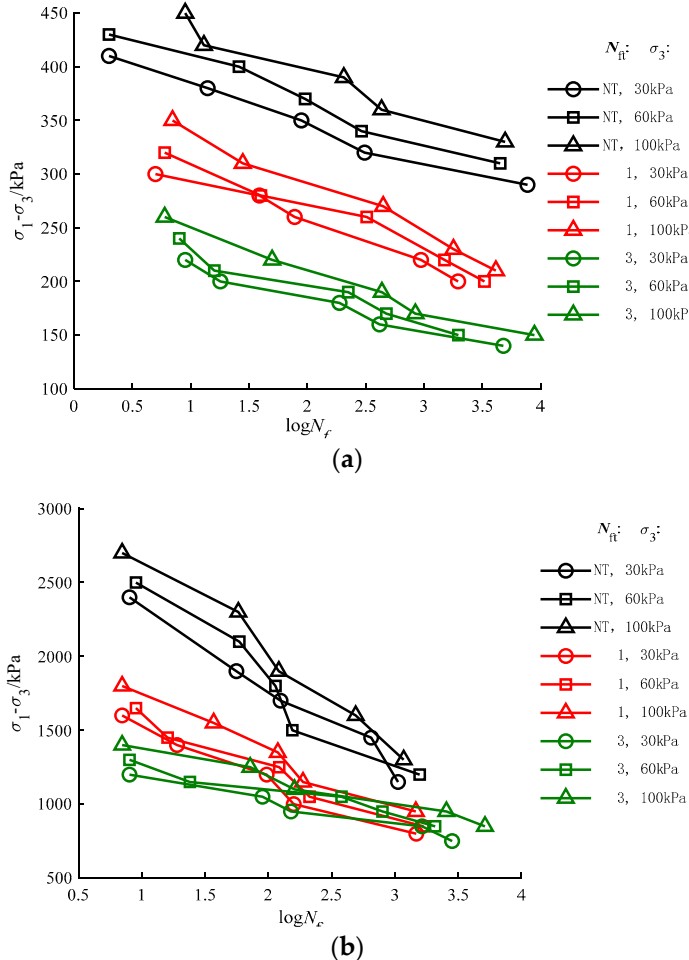

**Figure 1.** Dynamic strength curves of silty clay and cement-improved soil subjected to freeze-thaw cycles: (**a**) curves of plain filler when $N_{ft} = 0$, 1, and 3 under confining pressure of 30 kPa, 60 kPa, and 100 kPa; (**b**) curves of cement-improvement silty clay when $N_{ft} = 0$, 1, and 3 under confining pressure of 30 kPa, 60 kPa, and 100 kPa, where NT means normal temperature.

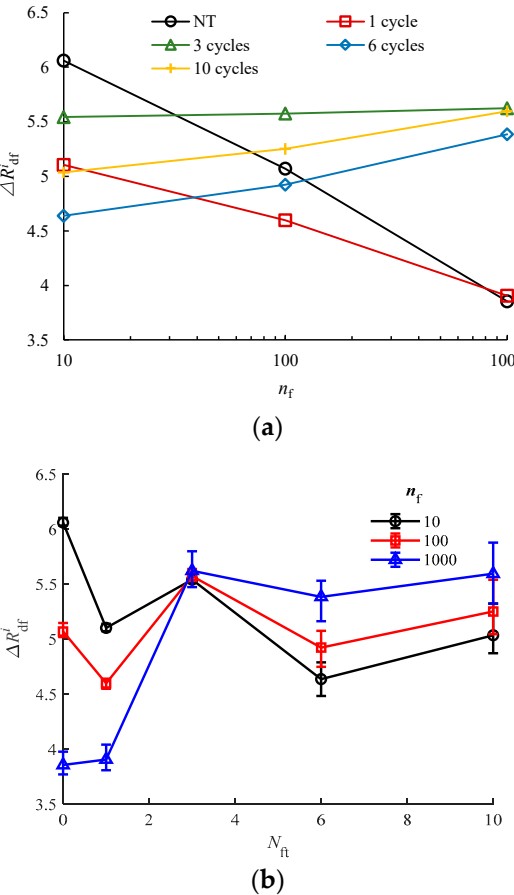

**Figure 2.** Relative increase in dynamic strength-impact factor relations: (**a**) variation of $\Delta R_{df}^i$ with $n_f$ at different $N_{ft}$; (**b**) variation of $\Delta R_{df}^i$ with $N_{ft}$ at different $n_f$.

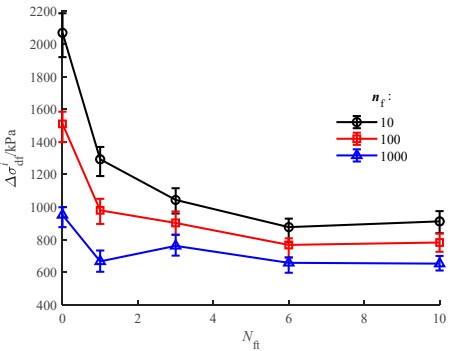

**Figure 3.** Variations of the absolute increase in dynamic strength of cement-improved soil with freeze-thaw cycles at different dynamic loading number.

## 3.2. Dynamic STRENGTH index

Based on the dynamic strength data obtained from the above tests, the $\sigma_{df}$-$\sigma_3$ relationship under different freeze-thaw cycles when $n_f$ = 10, 100, and 1000 was extracted and analyzed using linear regression analysis. The dynamic strength index under different working conditions was calculated using Mohr–Coulomb's law. Figures 4 and 5 show the $C_d$-$N_{ft}$ and $\varphi_d$-$N_{ft}$ relationship curves of soil mass before and after improvement. It can be seen from the figure that with the increase in freeze-thaw cycles, the $C_d$ and $\varphi_d$ of plain filler and cement-improved soil decrease in varying degrees, and the rate of decline decreases with the increase in freeze-thaw cycles. According to the variation trend $C_d$ of plain filler and $\varphi_d$ of cement-improved soil with $N_{ft}$, it can be seen that the dynamic strength indexes

tend to be stable after four and six cycles of freeze-thaw cycles, respectively. This is an indication that after many freeze-thaw cycles, the cohesion tends to be in equilibrium with the frost heave force. Compared to before freezing and thawing, the pores do not expand any further after freezing and thawing, and the internal structure of the soil reaches a dynamically stable state. When cement-improved soil is damaged by local cementation in the freeze-thaw cycle, $C_d$ decays more rapidly with $N_{ft}$, and more $N_{ft}$ is needed for stability. It is worth noting that it can be seen from Figure 4a that with the increase in $n_f$, the freeze-thaw cycle required for the $C_d$ of the cement-improved soil to reach the dynamic stable state is less, but this trend is not significant. It may be caused by an experimental error, which needs further experimental study.

　　　Figure 6 shows the variation curves of the relative increase in $\Delta R_C^i$ and $\Delta R_\varphi^i$ of $C_d$ and $\varphi_d$ with $N_{ft}$ when $n_f$ = 10, 100, and 1000, respectively. As shown in the figure, in general, $\Delta R_C^i$ is between 1.5 and 3 times; while $\Delta R_\varphi^i$ is between 1.5 and 4 times. The relationship between $n_f$ and $\Delta R_C^i$-$N_{ft}$ and its influence show a similar change rule to that of $\Delta R_{df}^i$. The change law of $\Delta R_\varphi^i$ is quite different from that of $\Delta R_c^i$, and $\Delta R_\varphi^i$ is not affected by $n_f$ and is relatively stable without freezing and thawing, After freezing and thawing, $\Delta R_\varphi^i$ increases with the increase in $n_f$. There is a peak point in the $\Delta R_\varphi^i$-$N_{ft}$ relationship and is inversely proportional to $n_f$ to reach the peak point. $\Delta R_\varphi^i$ increases with the increase in $N_{ft}$ and gradually decreases after the peak strain, and the rate of change of $\Delta R_\varphi^i$ with respect to $N_{ft}$ is also proportional to $n_f$. It can be seen that the cement improvement can effectively reduce the mechanical occlusion loss between soil particles after multiple freeze-thaw cycles, inhibit mutual movement, and have a better freeze-thaw resistance effect.

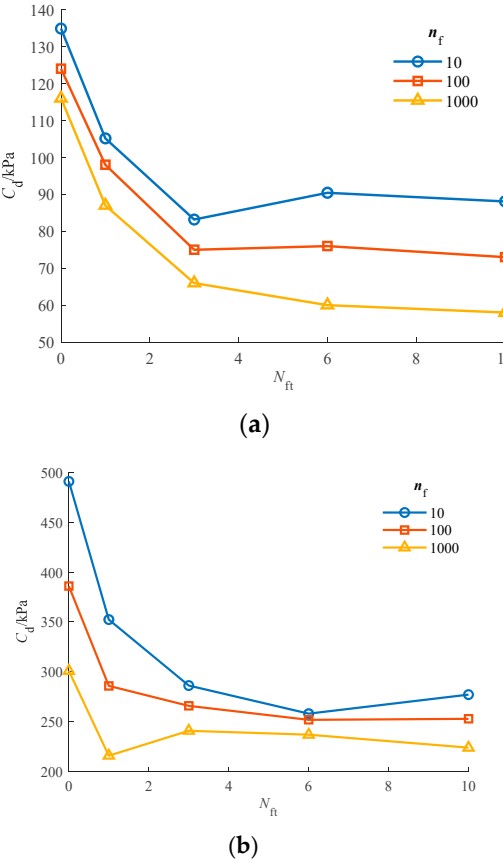

**(a)**

**(b)**

**Figure 4.** Variations of dynamic cohesion of soils with freeze-thaw cycles at different dynamic loading numbers: (**a**) curves of plain filler when $n_f$ = 10, 100, and 1000; (**b**) Curves of cement-improvement silty clay when $n_f$ = 10, 100, and 1000.

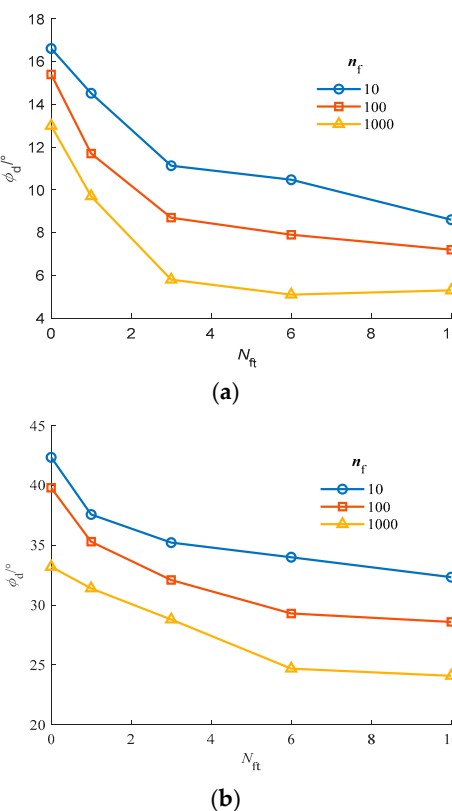

(a)

(b)

**Figure 5.** Variations of dynamic internal friction angle of soils with freeze-thaw cycles at different dynamic loading numbers: (**a**) curves of plain filler when $n_f$ = 10, 100, and 1000; (**b**) curves of cement-improved silty clay when $n_f$ = 10, 100, and 1000.

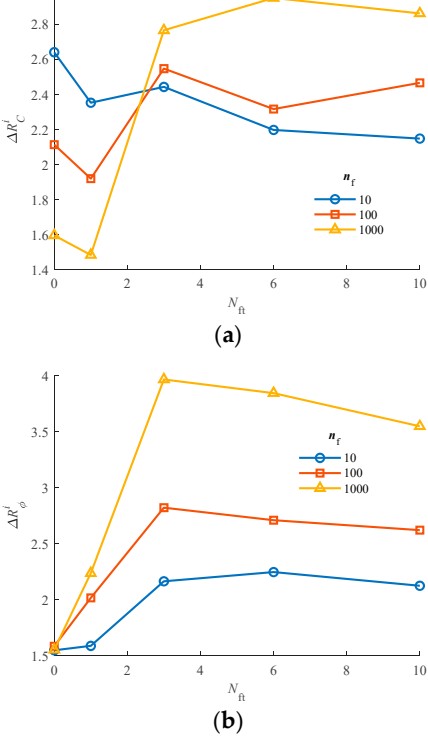

(a)

(b)

**Figure 6.** Variations of dynamic strength index of soils with freeze-thaw cycles at different dynamic loading numbers: (**a**) curves of $\Delta R_C^i$ when $n_f$ = 10, 100, and 1000; (**b**) curves of $\Delta R_\varphi^i$ when $n_f$ = 10, 100, and 1000.

*3.3. Critical Dynamic Stress*

The unrecoverable cumulative plastic strain of subgrade soil was continuously generated under the action of dynamic train-load cycles. According to whether the accumulative plastic strain reaches the failure standard under the action of train loads, the relationship between accumulative plastic strain $\varepsilon_p$ and dynamic loading number $n_f$ can be divided into two types: failure-type and attenuation-type. The dynamic stress level corresponding to the $\varepsilon_p$-n relationship between the stable and destructive types is the critical dynamic stress. China's Code for Design of Heavy-Haul Railway (TB 10625-2017) also takes the critical dynamic stress as the main indicator for the stability checking calculation of subgrade.

Through the definition of critical dynamic stress, it can be seen that its essence is the dynamic strength value of soil when the dynamic loading number tends to infinity. Obviously, it cannot be realized by the limited number of dynamic tests, and the critical dynamic stress value can only be approximated by statistical means. In this study, based on the critical dynamic stress ratio ($\sigma_{dcr}/2\sigma_3$)-confining pressure $\sigma_3$ power function relationship model proposed in reference [28], the maximum dynamic stress amplitude and the minimum dynamic stress amplitude corresponding to the attenuation-type and failure-type relationships under each confining pressure were extracted for a certain freeze-thaw cycle condition. The logarithmic linear regression of the dynamic stress amplitude-confining pressure data was used to estimate the critical dynamic stress of the subgrade filler.

Figure 7 shows the influence curve of freeze-thaw cycle on the critical dynamic stress of plain filler and cement-improved soil. During the change of confining pressure from 30 kPa to 100 kPa, the critical dynamic stress of unfrozen plain filler and cement-improved soil increased by 35 kPa and 140 kPa, respectively, while after 10 cycles of freeze-thaw cycles, the increment is reduced to 10 kPa and 70 kPa. It can be seen that the increase in confining pressure has a certain lifting effect on the critical dynamic stress of soil after freeze-thaw cycle, but the enhancement effect gradually weakens with the increase in freezing-thawing cycles.

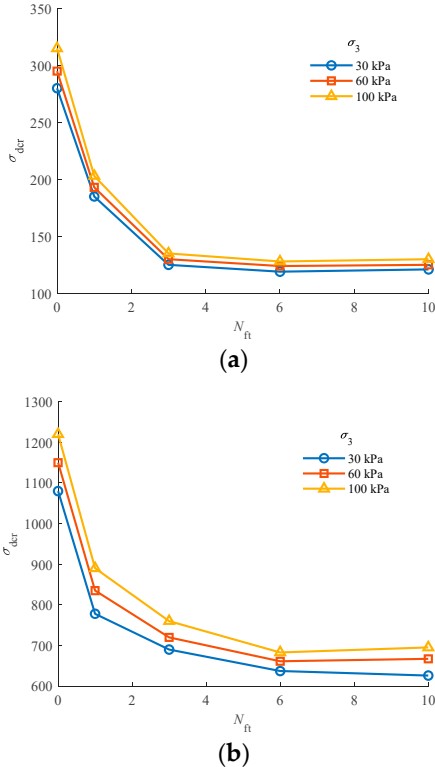

**Figure 7.** Variations of critical dynamic stress of soils with freeze-thaw cycles at different confining pressures: (**a**) curves of plain filler when $\sigma_3$ = 30, 50, and 100 kPa; (**b**) curves of cement-improved silty clay when $\sigma_3$ = 30, 50, and 100 kPa.

It can be seen from Figure 7 that after four freeze-thaw cycles, the critical dynamic stress of the plain filler tends not to change significantly with the number of freeze-thaw cycles. The critical dynamic stress of cement-improved soil also approximates to reach stability after six freeze-thaw cycles, which is basically consistent with the results of the literature [16] and the "Experimental Study on the Dynamic Characteristics of Improved Soil in Freeze-thaw Cycles of Qinhuangdao-Shenyang High-speed Railway" for cement-improved soil (six cycles and five cycles, respectively).

The variations of the mean values of critical dynamic stress difference $\Delta\sigma_{dcr}^i$ and the relative increase in critical dynamic stress $\Delta R_{dcr}^i$ with freeze-thaw cycles under different confining pressures are shown in Figure 8. It can be seen that both $\Delta\sigma_{dcr}^i$ and $\Delta R_{dcr}^i$ reach a dynamic steady state at around six cycles of freeze-thaw cycles, and the cement improvement has a significant effect on the critical dynamic stress of the silty clay. $\Delta\sigma_{dcr}^i$ decreases with the increase in freeze-thaw cycle, indicating that the increasing effect of cement improvement on the critical dynamic stress of silty clay decreases with the increase in the number of freeze-thaw cycles, where the attenuation rate of critical dynamic stress of cement-improved soil with the number of freeze-thaw cycles is greater than that of plain filler. In contrast, $\Delta R_{dcr}^i$ increases with the increase in freeze-thaw cycle, which indicates that the relative improvement effect of cement improvement on the critical dynamic stress of soil is significant after three cycles of freeze-thaw.

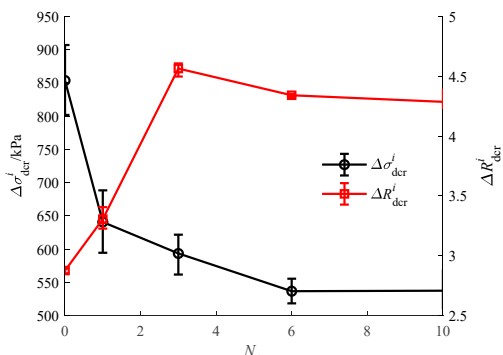

**Figure 8.** Variations of the mean values of critical dynamic stress difference and the relative increase in critical dynamic stress with freeze-thaw cycles under different confining pressures.

The critical dynamic stress attenuation coefficient $\eta_{dcr}$ of the freeze-thaw cycle was defined as the ratio of the critical dynamic stress after the freeze-thaw cycle to that of unfrozen soil, and the values of $\eta_{dcr}$ for plain filler and cement-improved soil under different freeze-thaw cycles are shown in Figure 9. It can be seen from the table that the confining pressure has little effect on $\eta_{dcr}$. With the increase in freeze-thaw cycles, the $\eta_{dcr}$ value of plain filler and cement-improved soil decreased continuously and finally tended to be stable, the average values were 0.41 and 0.57 after 10 cycles, respectively. Compared with the values of $\eta_{dcr}$ for the northeast clayey soil of D-group soil before and after cement improvement in the literature [16] under the same working conditions, the values of $\eta_{dcr}$ of silty clay in this study were 0.07 higher, while the values of $\eta_{dcr}$ of cement-improved soil decreased by 0.15. It can be seen that for different types of fillers, there are great differences in the values of $\eta_{dcr}$ between their own and improved soil. Dynamic stability tests under freeze-thaw cycles should be carried out according to specific soil characteristics. Figure 9 can provide a useful reference for the stability design of subgrade along the railway under the dynamic load of heavy-haul trains in seasonally frozen soil regions where silty clay is widely distributed in China.

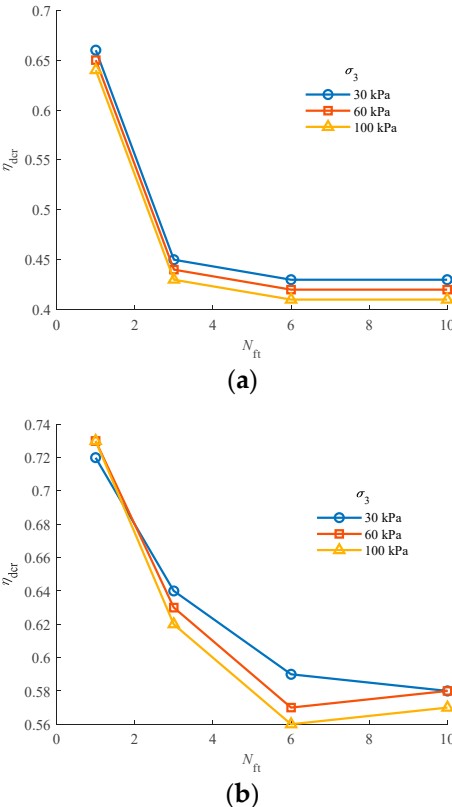

**Figure 9.** Variations of critical dynamic stress attenuation coefficient of soils with freeze-thaw cycles at different confining pressures: (**a**) curves of plain filler when $\sigma_3 = 30$, 50, and 100 kPa; (**b**) curves of cement-improved silty clay when $\sigma_3 = 30$, 50, and 100 kPa.

## 4. Conclusions

The dynamic strength characteristics of silty clay filler of heavy haul railway subgrade before and after the cement improvement under freeze-thaw cycle were studied by dynamic triaxial test, and the following conclusions are obtained:

(1) In the same environment, the dynamic strength of cement-improved soil was increased by 2.8~5.2 times, the dynamic cohesion was increased by 1.5~3 times, the dynamic internal friction angle was increased by 1.5~4 times, and the critical dynamic stress was increased by 2.9~4.3 times, compared with silty clay.

(2) The relative improvement effect of cement improvement on the dynamic strength parameters (dynamic strength, dynamic strength index, and critical dynamic stress) of silty clay is more significant after multiple freeze-thaw cycles and is less affected by confining pressure within the depth of subgrade bed.

(3) The dynamic strength parameters of the soil before and after improvement decrease with the increase in freeze-thaw cycles. The dynamic strength parameters of silty clay filler and cement-improved soil tend to be stable after four cycles and six cycles of freeze-thaw cycles, respectively.

(4) The critical dynamic stress attenuation coefficient $\eta_{dcr}$ was introduced to reflect the decay ratio of the critical dynamic stress before and after the freeze-thaw cycle. After 10 cycles of freeze-thaw, the average values of the plain filler and the cement-improved soil were 0.41 and 0.57, respectively. For different types of cohesive soils, there were great differences in their own values and the improved values of $\eta_{dcr}$, so the dynamic stability test under the freeze-thaw cycle should be conducted according to the specific type of soil.

(5) In the seasonal permafrost zone, cement-improved silty clay can be used in heavy-duty railroad engineering applications with reference to the degradation regulations

of dynamic strength parameters proposed in this paper for subgrade design and subgrade dynamic stability inspection and testing cycles.

**Author Contributions:** Conceptualization, methodology, validation, investigation, resources and writing—original draft preparation, Z.M. and Z.X.; writing—review and editing, Y.H.; supervision and funding acquisition, Y.Z. All authors have read and agreed to the published version of the manuscript.

**Funding:** This research was funded by the Central Guidance for Local Science and Technology Development Funds Project, grant number 2022ZY0052; the Open Research Fund Program of the State Key Laboratory of Frozen Soil Engineering of China, grant number SKLFSE201907; the Basic Research Projects of Heilongjiang Provincial Higher Education Institutions, grant number 2020-KYYWF-0269; the Science and Technology Plan Project of Zhejiang Provincial Department of Transportation, grant number 202320-2 and the Doctoral natural science foundation of East China University of Technology, grant number DHBK2019243. The APC was funded by 2022ZY0052.

**Institutional Review Board Statement:** Not applicable.

**Informed Consent Statement:** Not applicable.

**Data Availability Statement:** The data presented in this study are available on request from the corresponding author.

**Acknowledgments:** The materials used for experiments were provided by Qingdao Panyao New Material Engineering Research Institute Co., Qingdao, China. The authors are grateful to Ling for offering the detailed parameters for the test materials in this study.

**Conflicts of Interest:** The authors declare no conflict of interest.

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
