# Peer review of "Dynamic Strength Characteristics of Cement-Improved Silty Clay under the Effect of Freeze-Thaw Cycles"

_sustainability, doi:10.3390/su15043333_

Round 1

Reviewer 1 Report

Authors have contributed towards Freeze-thaw Cycles, in an innovative manner. Therefore few comments required to be addressed.

1] Abstract needs more modifications and improvement.

2] Add the major contributions in the introduction.

3] more rigorous literature survey is required.

4.] add more recent literatures.

5] enrich the results and analysis.

6] Table 1 needs more improvement. 

7] Fig. 1 needs more explanation.

8] After section 1, directly authors have included the section 2 as experimental scheme, which is not acceptable. Include the system description or any method that authors want to justify. Add mathematical modeling.

Reviewer 2 Report

In this manuscript, a series of dynamic triaxial tests were conducted to study the dynamic strength characteristics of silty clay filler before and after cement improvement by freeze-thaw cycles. This research is interesting, and the manuscript is also well written. However, there are a few minor problems that need to be corrected before this manuscript can be accepted.

(1)    In the introduction, the author should clearly point out the innovation of this research.

(2)    In lines 66-68: ‘The test soil was taken from a soil material landfill in Beijing-Harbin high-speed railway, and the tested soil was airdried, crushed using a rubber pestle, and the fillers were sieved through a 2 mm sieve.’ Why 2 mm? Is there any basis for that?

(3)    The format of text in Table 1 needs to be edited again.

(4)    In lines 88-89: ‘The height of the sample was 125mm and the diameter was 61.8mm.’ Why is the specimen set to this size. Similarly, in Section 2.3, the setting of many test parameters requires the provision of corresponding standards or other scholars' research results, otherwise these test parameters are unfounded.

(5)    In lines 140-142: ‘It can be seen from the figure that the dynamic strength of plain filler and cement-improved soil decreases significantly after freeze-thaw cycles, and decreased continuously with the increase of dynamic loading number.’ The author needs to explain this experimental phenomenon briefly. The research results of Wang et al. (2021, 2022) show that the meso-fabric of geomaterials has a significant influence on its strength. The increase of dynamic loading number destroys the meso-fabric of plain filler and cement improved soil, which may be the main reason for this result.

Strength and dilatancy of coral sand in the South China Sea. Bulletin of Engineering Geology and the Environment. 2021, 80 (10): 8279–8299. DOI: 10.1007/s10064-021-02348-6.

Particle size and confining-pressure effects of shear characteristics of coral sand: an experimental study. Bulletin of Engineering Geology and the Environment. 2022, 81 (3): 97. DOI: 10.1007/s10064-022-02599-x.

(6)    In Section 3.3, what is the most important conclusion obtained by the author based on the critical dynamic stress?

(7)    In the conclusion section, can you add some enlightenment to engineering practice based on the test results?

Reviewer 3 Report

This research refers to the variation law of dynamic strength of cement modified silty clay under freeze-thaw cycle by dynamic triaxial test.

Overall the article is good and nicely written and it brings some interesting and useful information. I think that the manuscript has a potential to be published in the journal.However, I believe that the manuscript can be improved further. Thus I recommend the revising of the article based on the main points mentioned below.

-        The novelty of the study is not properly highlighted in the introduction section.

-        Some phrases are too long, making it difficult to understand the idea.

-        Obtained results need to be compared with other reports to underline the performance.

-        More recent publications, especially in the last 10 years, should be added to your literature.

 Other comments are scored on article text.

Round 2

Reviewer 1 Report

no comments